# Clinical Characteristics of Patients with De Novo Parkinson’s Disease and a Positive Family History

**DOI:** 10.3390/medicina60091378

**Published:** 2024-08-23

**Authors:** Hyunsoo Kim, Seong-Min Choi, Hak-Loh Lee, Soo Hyun Cho, Byeong C. Kim

**Affiliations:** 1Department of Neurology, Chonnam National University Hospital, Gwangju 61469, Republic of Korea; kharistma@naver.com (H.K.); ss10004ok@naver.com (H.-L.L.); k906141h@hanmail.net (S.H.C.); byeong.kim7@gmail.com (B.C.K.); 2Department of Neurology, Chonnam National University Medical School, Gwangju 61469, Republic of Korea

**Keywords:** Parkinson’s disease, family history, motor symptoms, non-motor symptoms

## Abstract

*Background/Objectives*: A family history of Parkinson’s disease (PD) is an important risk factor for developing PD. Because only a few studies have investigated the clinical characteristics of PD patients based on family history, this study compared the clinical characteristics of PD patients with and without a family history of PD. *Methods*: The study involved 356 patients with de novo PD. The data on the patients’ PD family histories were obtained from the patients and their caregivers. Motor and non-motor PD symptoms were assessed using the appropriate scales. *Results*: Out of the 356 PD patients, 26 (7.3%) had a family history of PD. Compared with patients without a family history of PD, those with a family history of PD tended to be younger at diagnosis (67.9 years vs. 62.2 years, respectively; *p* = 0.009) and exhibited significantly more severe rigidity (*p* = 0.036). Motor subtype was not different between the PD patients with and without a family history. PD patients with a family history experienced significantly fewer falls/cardiovascular symptoms within the Non-Motor Symptoms Scale domains (*p* = 0.001) compared to their counterparts, although this was not statistically significant upon adjusting for age (*p* = 0.119). *Conclusions*: In de novo PD patients, having a family history of PD is associated with a younger age at diagnosis and more severe rigidity.

## 1. Introduction

Parkinson’s disease (PD) is a clinically heterogeneous neurodegenerative disorder, presenting with cardinal symptoms of parkinsonism including rest tremor, bradykinesia, rigidity, and gait disturbance [1]. PD patients may also develop various non-motor symptoms, such as cognitive decline, depression, autonomic dysfunction, and sleep disorders [2,3]. Numerous studies have investigated and identified the various risk factors underlying PD [4,5,6]. Genetic factors are strongly associated with PD and can interact with environmental factors to increase the risk of PD [7,8,9,10].

A family history of PD, one of the strong risk factors for developing PD [9,11,12], is reported in about 15% of PD patients [7]. Familial aggregation studies reported an increased risk of PD amongst first-degree relatives, supporting the importance of genetic factors in the etiology of PD [13]. Several genes associated with PD have been identified, and they are known to contribute to the development and progression of the disease by disrupting interconnected processes such as protein aggregation, mitochondrial dysfunction, and cell migration, which are key mechanisms in PD pathogenesis [3]. There is an ongoing debate regarding the impact of the family history of PD on the demographic and clinical features of PD patients. Patients with familial PD exhibit early disease onset [12]. Motor features of patients with familial PD do not differ from those of patients with non-familial PD [12], while the differences in non-motor features between them are significant [14,15,16].

Although the family history of PD is an important risk factor, few studies compared the clinical characteristics of PD patients with and without a positive family history [12]. Thus, this study compared the clinical characteristics of PD patients with and without a positive family history.

## 2. Materials and Methods

### 2.1. Study Population and Collection of Clinical Data

We enrolled patients with de novo PD who visited our outpatient clinic from January 2018 to December 2021. None of them were taking dopaminergic medications during the clinical evaluations. PD diagnosis was made according to the clinical diagnostic criteria of the United Kingdom Parkinson’s Disease Society Brain Bank [17]. All included patients were followed up at 3-month intervals for at least 1 year to confirm the response to dopaminergic medication. Patients who exhibited a poor response to dopaminergic medications or had features of Parkinson-plus syndrome, secondary parkinsonism, or any significant brain lesions on magnetic resonance imaging associated with parkinsonism were excluded from this study. All participants underwent a detailed neurologic examination by a trained neurologist. The institutional review board of our hospital approved this study, which was performed according to the ethical standards of the 1964 Declaration of Helsinki.

All clinical evaluations, including a parkinsonism assessment, were performed before starting the anti-Parkinson medication. Using a questionnaire, we identified whether the first-degree relatives (siblings and patients) of the PD patients had PD. Those with PD in second-degree relatives were excluded. Information on age, sex, age at symptom onset, disease duration, and level of education was obtained from each patient. The motor and non-motor symptoms of PD were evaluated using the scales such as, the Unified Parkinson’s Disease Rating Scale (UPDRS), the modified Hoehn–Yahr (mHY) stage, the Non-Motor Symptoms Scale (NMSS), the Mini-Mental Status Examination (MMSE), and the Beck Depression Inventory (BDI) [18,19,20,21,22]. Based on previous studies that classified PD patients with motor symptoms, the patients were grouped into the tremor-dominant subtype, the mixed subtype, and the akinetic-rigid subtype [23]. Then, we investigated family history differences across these subtypes.

### 2.2. Data Analysis

Variables were expressed as mean ± standard deviation. Student’s *t*-tests and chi-squared tests were used to compare the statistical differences between the PD clinical characteristics in patients with and without a positive family history. Because PD patients with a positive family history were younger at diagnosis than those without a family history, we performed an analysis of covariance (ANCOVA) to control for age differences at diagnosis. All significance tests were two-tailed. A *p* < 0.05 indicated statistically significant differences. Statistical analyses were performed using SPSS version 27.0 (IBM Corp.; Armonk, NY, USA).

## 3. Results

Of the 356 PD patients included in this study, 26 (7.3%) had a family history of PD in their first-degree relatives. Table 1 summarizes the clinical characteristics of the patient groups based on their family history of PD. PD patients with a positive family history were significantly younger at diagnosis when compared with their counterparts (62.2 years vs. 67.6 years, *p* = 0.009). There were no significant differences between the two groups regarding other clinical characteristics, such as sex, education level, age at symptom onset, duration of disease, mHY stage, UPDRS III and II scores, NMSS total score, MMSE score, and BDI score.

Table 2 and Figure 1 compare Parkinsonian motor features in patients with and without a family history of PD. The severity of rigidity was significantly higher in PD patients with a positive family history than in patients with a negative family history (*p* = 0.036), even after adjusting for age differences (*p* = 0.032). However, other motor symptoms, such as tremor, bradykinesia, and gait and posture disturbances did not differ significantly between the two groups.

The analysis of motor subtype differences between the two groups revealed that all three subtypes did not differ significantly (Table 3).

A comparison of each NMSS domain revealed that PD patients with a positive family history had significantly fewer falls/cardiovascular symptoms than their counterparts (*p* = 0.001), although these differences were not statistically significant after adjusting for age differences (*p* = 0.119). The two groups did not differ significantly in other NMSS domains (Table 4).

## 4. Discussion

Our analysis showed that patients with PD and a positive family history were significantly younger at diagnosis and had higher rigidity scores compared with patients with PD and a negative family history. However, non-motor symptoms did not differ significantly between the two groups.

In this study, the proportion of patients with a history of PD in first-degree relatives was 7.3%, which is lower compared to other studies [24]. The reported proportion of PD patients with a history of PD in first-degree relatives was 10–20%, with PD patients being 2–3 times more likely to have a family history of PD than controls [12]. The low rate of PD family history in our study might be caused by recall bias since we did not investigate the medical records of first-degree relatives. A meta-analysis found that individuals with a family history of PD had a 4.45 times higher risk of developing PD when compared with those without a family history, regardless of the degree of family relationship. However, another study found that the relative risk was slightly lower (3.23 times) in individuals with a history of PD in first-degree relatives [25]. Such variability in reported relative risks might stem from differences in the interactions between genetic and environmental risk factors.

Previous studies consistently showed that individuals with a family history of PD tend to be younger at PD onset when compared with those without a family history [26,27,28,29,30]. Our study also showed that patients with a family history of PD were younger at PD symptom onset than those with a negative family history (62.8 years vs. 65.9 years), although the difference was not statistically significant. Meanwhile, we found that patients with a family history of PD were younger at PD diagnosis compared to their counterparts. Therefore, people with a family history of PD seem to visit the hospital sooner when Parkinson’s symptoms occur compared to those with a negative family history. In our study, the family history of PD did not differ by sex. Although some studies reported that more female patients have a family history of PD than men [24,31], others did not find differences in the family history rates of PD between men and women [32].

Research on the association between family history and the motor subtypes of PD patients is limited. Our analysis did not find an association between family history and PD motor subtypes. However, previous reports have shown that a family history of PD was a significant risk factor for the development of PD only in the tremor-dominant subtype and the akinetic-rigid subtype with earlier onset [33]. Familial and sporadic PD does not exhibit differences in motor features, such as bradykinesia, tremor, and rigidity [34]. However, our data demonstrated that patients with a family history of PD had higher rigidity scores than their counterparts. Consistent with our findings, patients with early-onset PD, which is more likely to be associated with a family history of PD, exhibit rigidity as the predominant initial symptom more often than patients with middle- and late-onset PD [35]. Furthermore, the results of our study might be meaningful because patients with a positive family history had higher rigidity scores despite the shorter disease duration when compared with those without a family history. Genetic research on patients with familial PD has provided valuable insights into the underlying genetic mutations and pathogenic mechanisms of the disease. Identifying genetic mutations early can offer important information for prognosis and potential targeted treatments. In clinical practice, younger patients diagnosed with PD, especially those exhibiting severe rigidity, are more likely to have a familial form of the disease. Therefore, it is recommended that more comprehensive genetic testing be conducted for such patients.

The differences in non-motor symptoms between familial PD and sporadic PD are diverse [14,15,16]. The prevalence of sleep disturbances is lower in patients with familial PD than in those with sporadic PD [14], but the degree of olfactory dysfunction does not differ between these PD types [15]. However, some reports have indicated that a family history of PD is associated with more severe cognitive function, depression, and anxiety based on NMSS assessment [16]. A family history of PD could also accelerate MMSE progression [31]. In our study, we found that cardiovascular problems were less common in patients with a family history of PD, although the difference was not significant after correcting for age differences. Thus, there are no clear differences in NMSS subdomains between patients with and without a family history of PD.

Our study provides valuable insights by comparing the demographic features and motor and non-motor symptoms of patients with and without a family history of PD, which has not been sufficiently investigated in previous studies. However, our study has several limitations to consider. First, the retrospective, single-center design may limit the generalizability of our findings by introducing patient selection bias and other data issues inherent to retrospective research, making it difficult to apply our results to a broader population. Second, the study relied on the patient’s self-reported family history of PD, and genetic testing was not performed to identify the genetic lesions underlying the disease. Third, because we did not include a control group, we could not compare family history rates in individuals with and without PD. Finally, the interpretation of the results might be affected by differences in environmental factors, which we did not account for.

## 5. Conclusions

Our study showed that PD patients with a family history were significantly younger at diagnosis and exhibited more severe rigidity when compared with those without a family history of PD. Therefore, if a PD patient is young or has severe rigidity at the time of diagnosis, more comprehensive genetic testing may be necessary. Because this was a small single-center study, larger follow-up studies are warranted.

## Figures and Tables

**Figure 1 medicina-60-01378-f001:**
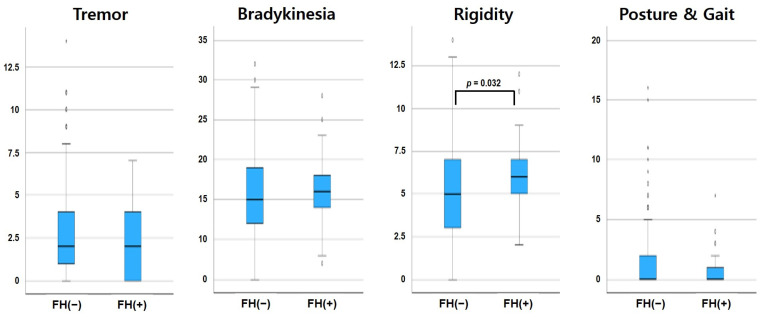
A box plot for the comparison of parkinsonian symptoms in patients with and without a family history of Parkinson’s disease.

**Table 1 medicina-60-01378-t001:** Clinical characteristics of Parkinson’s disease patients with and without a family history of Parkinson’s disease.

	All Patients (n = 356)	FH Negative (n = 330)	FH Positive (n = 26)	*p*-Value
Age (years)	67.2 ± 10.3	67.6 ± 9.6	62.2 ± 15.9	0.009
Sex (female:male)	210:146	196:134	14:12	0.580
Education (years)	8.9 ± 4.8	8.8 ± 4.8	10.2 ± 4.3	0.152
Age at onset (years)	65.6 ± 9.7	65.9 ± 9.7	62.8 ± 10.2	0.128
Duration of disease (months)	20.1 ± 22.3	20.4 ± 22.7	16.4 ± 16.2	0.373
mHY stage	1.9 ± 0.7	1.9 ± 0.7	2.0 ± 0.5	0.584
UPDRS part III score	21.8 ± 10.5	21.8 ± 10.7	22.5 ± 8.1	0.751
UPDRS part II score	7.1 ± 6.3	7.1 ± 6.4	6.0 ± 5.2	0.389
NMSS total score	48.3 ± 33.3	48.3 ± 33.5	48.2 ± 31.1	0.989
MMSE	26.0 ± 3.7	25.9 ± 3.8	26.6 ± 2.6	0.358
BDI	11.6 ± 10.0	11.7 ± 10.1	10.7 ± 8.2	0.636

Values represent mean ± standard deviation. FH, family history; mHY, modified Hoehn-Yahr; UPDRS, Unified Parkinson’s Disease Rating Scale; NMSS, Non-Motor Symptoms Scale for Parkinson’s disease; MMSE, Mini-Mental Status Examination; BDI, Beck Depression Inventory.

**Table 2 medicina-60-01378-t002:** Comparison of parkinsonian symptoms in patients with and without a family history of Parkinson’s disease based on the Unified Parkinson’s Disease Rating Scale.

	FH Negative (n = 330)	FH Positive (n = 26)	*p*-Value
Tremor	2.7 ± 2.5	2.4 ± 2.1	0.549
Rest tremor	1.7 ± 1.9	1.3 ± 1.4	0.308
Action tremor	1.0 ± 1.2	1.1 ± 1.3	0.724
Bradykinesia	10.6 ± 5.3	11.1 ± 4.1	0.696
Rigidity	5.0 ± 2.9	6.1 ± 2.4	0.036 (0.032 *)
Gait and Posture	1.4 ± 2.3	1.0 ± 1.6	0.396

Values represent mean ± standard deviation. FH, family history. * Age was included as the ANCOVA covariate.

**Table 3 medicina-60-01378-t003:** Motor subtypes of Parkinson’s disease patients with and without a family history of Parkinson’s disease.

	FH Negative (n = 330)	FH Positive (n = 26)	*p*-Value
Tremor-dominant	100	6	0.722
Mixed	35	3	
Akinetic-rigid	194	17	

FH, family history.

**Table 4 medicina-60-01378-t004:** Comparison of the Non-Motor Symptoms Scale subdomains in Parkinson’s disease patients with and without a family history of Parkinson’s disease.

	FH Negative (n = 330)	FH Positive (n = 26)	*p*-Value
D1 falls/cardiovascular	2.5 ± 3.9	1.1 ± 1.7	0.001 (0.119 *)
D2 sleep/fatigue	9.3 ± 8.0	8.1 ± 8.8	0.496
D3 mood/cognition	7.6 ± 8.2	10.5 ± 9.9	0.086
D4 perception/hallucination	0.2 ± 1.0	0.3 ± 1.2	0.565
D5 attention/memory	3.9 ± 4.8	3.3 ± 4.0	0.571
D6 gastrointestinal	5.7 ± 6.1	5.5 ± 5.7	0.853
D7 urinary	14.8 ± 12.6	15.6 ± 14.2	0.764
D8 sexual	0.5 ± 1.6	0.3 ± 1.5	0.566
D9 others	3.6 ± 4.9	3.2 ± 3.1	0.708

Values represent mean ± standard deviation. FH, family history. * Age was included as the ANCOVA covariate.

## Data Availability

Anonymized data are available upon request from the corresponding author.

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
