# Peer review of "Clinical Characteristics of Patients with De Novo Parkinson’s Disease and a Positive Family History"

_medicina, 2024, doi:10.3390/medicina60091378_

Round 1

Reviewer 1 Report

Comments and Suggestions for Authors

The manuscript titled “Clinical characteristics of patients with de novo Parkinson's disease with a positive family history" explores a crucial aspect of Parkinson's disease (PD) regarding the impact of a positive family history on the clinical characteristics of de novo PD patients. This study finds that Parkinson's disease patients with a family history are diagnosed at a younger age and tend to show more rigidity in comparison to those without a family history. However, a few minor revisions are required to enhance the overall quality of the manuscript.

1-     Explain in detail the assessment criteria for ‘family history’. For example, if it was self-reported or if any earlier medical records were checked.

2-     For a better interpretation of the results, I suggest the authors make a figure that highlights the key findings of the manuscript using the data from the tables. For example, a pie chart or a box plot.

3-     Briefly discuss different genetic mutations or genetic pathways which are highly prevalent in familial forms of PD and link them with your findings.

4-     Discuss the clinical relevance of your findings, especially if your results have any potential to impact existing treatment strategies.

5-   Proofread the manuscript to avoid a few minor grammatical errors.

6-     Briefly expand the conclusion section and add a line or two to highlight the implications of your findings for clinical practice.

Comments on the Quality of English Language

The Quality of the English Language is fine.

Reviewer 2 Report

Comments and Suggestions for Authors

       The introduction discusses the role of family history in PD but doesn’t delve into potential genetic or environmental mechanisms. Please provide more detail on how genetic factors associated with family history might contribute to the development and progression of PD.

       The introduction mentions non-motor symptoms but doesn’t explore why these are important or how they might differ in familial versus sporadic PD. Does this study fill? How do your findings advance understand of PD, particularly in patients with a family history?

       The study’s single-center design should be more explicitly discussed in the limitations section.

       The methods state that patients were "followed up for at least 1 year," but there is no information on what this follow-up entailed. Were there specific intervals for follow-up visits? What outcomes were monitored during this period.

       While the study excludes patients with "Parkinson-plus syndrome, secondary parkinsonism, or any significant brain lesions on magnetic resonance imaging," the criteria for what constitutes "significant brain lesions" are not defined.

       In the introduction, it is mentioned that "a family history of PD is one of the strong risk factors," and later this is reiterated with similar information, stating "the relative risk of developing PD in first-degree relatives." Please consider removing any unnecessary repetition.
